# Characterization of Regulatory and Transporter Genes in the Biosynthesis of Anti-Tuberculosis Ilamycins and Production in a Heterologous Host

**DOI:** 10.3390/md18040216

**Published:** 2020-04-17

**Authors:** Jianqiao He, Xin Wei, Zhijie Yang, Yan Li, Jianhua Ju, Junying Ma

**Affiliations:** 1CAS Key Laboratory of Tropical Marine Bio-Resources and Ecology, Guangdong Key Laboratory of Marine Materia Medica, RNAM Center for Marine Microbiology, South China Sea Institute of Oceanology, Chinese Academy of Sciences, Guangzhou 510301, China; 13409727100@163.com (J.H.); weixin1011@126.com (X.W.); yzjie6@126.com (Z.Y.); ly505479742@126.com (Y.L.); jju@scsio.ac.cn (J.J.); 2University of Chinese Academy of Sciences, Beijing 100049, China; 3Key Special Project for Introduced Talents Team of Southern Marine Science and Engineering Guangdong Laboratory, Guangzhou 510301, China

**Keywords:** marine-derived *Streptomyces*, ilamycins, regulator, ABC-transporter, heterologous expression

## Abstract

Ilamycins are cyclopeptides with novel structures that have been isolated from different *Actinomycetes*. They showed strong anti-tuberculosis activity and could serve as important anti-tuberculosis drug leads. The functions of the pre-tailoring and the post-tailoring genes in the biosynthesis of ilamycins have been elucidated, but the functions of the regulatory and transporter genes remain elusive. We reported herein the functions of four genes in ilamycin biosynthetic gene cluster (*ila* BGC) including two regulatory genes (*ilaA* and *ilaB*) and two transporter genes (*ilaJ* and *ilaK*) and the heterologous expression of *ila* BGC. The IlaA and IlaB were unambiguously shown to be negative and positive regulator of ilamycins biosynthesis, respectively. Consistent with these roles, inactivation of *ilaA* and *ilaB* (independent of each other) was shown to enhance and abolish the production of ilamycins, respectively. Total yields of ilamycins were enhanced 3.0-fold and 1.9-fold by inactivation of *ilaA* and overexpression of *ilaB* compared to those of in the *Streptomyces atratus* SCSIO ZH16, respectively. In addition, the *ila* BGC was successfully expressed in *Streptomyces coelicolor* M1152, which indicated that all biosynthetic elements for the construction of ilamycins were included in the PAC7A6. These results not only lay a foundation for further exploration of ilamycins, but also provide the genetic elements for synthetic biology.

## 1. Introduction

Marine derived *Streptomyces* have been identified as sources for novel natural products that are of great interest to drug discovery and development [1,2,3]. Non-ribosomal peptides are an important class of natural products known for their structural diversity and bioactivity; many antitumor and anti-infective drugs are non-ribosomal peptides [4]. For example, romidepsin, used to treat T-cell lymphoma [5,6,7], daptomycin and vancomycin, used to treat multidrug-resistant infections [8,9,10,11,12], and capreomycin, used for tuberculosis (TB) [13], are all non-ribosomal peptides. Ilamycins, also known as rufomycins, are potent anti-TB cyclic heptapeptides containing three rare building blocks, l-3-nitrotyrosine, l-2-amino-4-hexenoic acid, and l-tryptophan, in which the indole nitrogen is alkylated with an isoprene or a modified isoprene unit. Ilamycins were cyclopeptides isolated from several strains [14,15,16]. Recently, eight more ilamycin analogs, ilamycins B_1_ (**1**), B_2_ (**2**), C_1_ (**3**), C_2_ (**4**), D (**5**), E_1_ (**6**), E_2_, and F (Figure 1A), were isolated by our group from a marine *Streptomyces atratus* SCSIO ZH16 and its engineered strain [17]. Bioactivity studies revealed that ilamycins, except ilamycin B_1_, have potent anti-TB activities [17,18], and that ilamycin E has the strongest anti-TB activity at a minimal inhibition concentration of 9.8 nM [17]. Anti-TB mechanistic studies revealed that these ilamycins are bactericidal to *Mycobacterium tuberculosis* through their inhibition of caseinolytic protease complex component (ClpC1) and subsequent modulation of the degradation of intracellular proteins [19]. Ilamycin C and ilamycin E have selective anti-triple-negative breast cancer activity and have been found to act on the interleukin-6/signal transducer and activator of transcription 3 (IL-6/STAT3) and endoplasmic reticulum stress-CCAAT-enhancer-binding protein homologous protein-B-cell lymphoma 2 (ER stress-CHOP-Bcl2) signaling pathways, respectively [20,21]. The biosynthesis of the rare l-3-nitrotyrosine and l-2-amino-4-hexenoic acid units as well as two post-tailoring steps in the ilamycin biosynthetic pathway have been studied by our group and others through a series of gene inactivation, precursor chemical complementation, isotope-labeled precursor feeding experiments as well as structural elucidation of three intermediates from the respective mutants [17,22]. However, the functions of the regulatory and transporter genes in the biosynthesis of these compounds in these *Streptomyces* strains have not been characterized. 

Herein, we aimed to identify and characterize the functions of the regulators and transporters involved in the biosynthesis of ilamycins, further use them to construct high-yield strains, and express the biosynthetic gene cluster of ilamycins in *Streptomyces coelicolor* M1152. 

## 2. Results

### 2.1. Identification of IlaA as a LysR Family Regulator and IlaB as a Streptomycin Biosynthesis Operon Regulator by Bioinformatics Analysis

The biosynthetic gene cluster of ilamycins consisted of 20 genes (Figure 1B), which had been identified in our previous study [17]. IlaA and IlaB were resided in the upstream of ilamycin biosynthetic gene cluster (*ila*). IlaA was firstly annotated as a LysR family transcriptional regulator. However, our analysis using the basic local alignment search tool (BLAST) revealed that it was annotated as a hypothetical protein, and only one homologue sequence derived from *Liparis tanakae* was found, which means its function remains elusive.

IlaB was annotated as a streptomycin biosynthesis operon regulator by BLAST analysis. To understand the sequence characteristics of IlaB, a sequence alignment of IlaB and four other homologs, DtpR2 [23], StaQ [24], StrR [25], and Veg1 [26], was performed with Clustal Omega [27], and the alignment result was viewed in ESPript [28], as shown in Figure 2. The online Pfam database was used to analyze the domains of streptomycin biosynthesis operon regulator [29]. The characteristics of streptomycin biosynthesis operon regulator are as follows: (i) it contains a ParB-like nuclease domain in the N-terminal of IlaB (49–125 amino acids for IlaB), which is involved in many cellular processes, such as cell cycle control, cell division, sporulation, and chromosome partitioning; (ii) there is a homeodomain-like domain in the middle of IlaB (137–173 amino acids for IlaB), which possesses multiple regulatory functions such as stress response, metabolism, transport/transfer, or regulation; (iii) a winged helix-turn-helix domain similar to that seen in AsnC/LacI family regulators resides almost the middle of IlaB (205–238 amino acids for IlaB).

### 2.2. Identification of IlaA and IlaB as Negative and Positive Regulators Regulating the Production of Ilamycins 

To investigate the functions of IlaA and IlaB in the biosynthesis of ilamycins, *ΔilaA* and *ΔilaB* mutants were constructed by the methods described previously [17]. The phenotype and the genotype were confirmed by (apramycin-resistance/kanamycin-sensitivity) Apr^R^/Kan^S^ and polymerase chain reaction (PCR) analyses. The PCR results are shown in Appendix A. In order to characterize the production of ilamycins in these two *ΔilaA* and *ΔilaB* mutants, the mutants of *ΔilaA* and *ΔilaB* were fermented under the conditions described in our previous studies [17], with the wild-type as a control. High-performance liquid chromatography (HPLC) analysis of the fermentation media extracts revealed that *ΔilaA* extracts produced higher titers of ilamycins (Figure 3, trace ii) than those of the wild type (Figure 3, trace i). The total yields of ilamycins produced by the *ΔilaA* mutants were enhanced 3.0-fold (11.76 mg/mL) when compared to those in the wild type (3.92 mg/mL) (Figure 3B). However, *ΔilaB* completely abolished the production of ilamycins (Figure 3, trace iii). These results indicate that IlaA and IlaB might work as a negative regulator and a positive regulator, respectively, in the biosynthesis of ilamycins. 

To verify whether the abrogation of ilamycin production in *ΔilaB* mutant was indeed caused by the deletion of *ilaB*, a complementation experiment to *ΔilaB* mutant was performed. The *ilaB* gene was expressed under the control of *PermE** in pL646ATE vector in *Escherichia coli* ET12567/pUZ8002, which was introduced into *ΔilaB* mutant by inter-genetic conjugation. The complement mutants *ΔilaB::ilaB*-pL646ATE were fermented under standard conditions for ilamycin production. HPLC analysis showed that the production of ilamycins was restored in the complement mutant (Figure 3, trace iv) as those seen in the wild type. Taken together, these results conclusively establish the positive regulatory role of IlaB in the production of ilamycins. 

### 2.3. Overexpression of IlaB Enhanced the Production of Ilamycins

Positive regulators play important roles in the biosynthesis of secondary metabolites, and their overexpression can directly improve the titers of these secondary metabolites. To examine whether IlaB, as a positive regulator, could increase the yields of ilamycins, overexpression plasmids were constructed under the control of the *PermE** in the pL646ATE vector and then introduced into *S. atratus* SCSIO ZH16 to generate *ilaB* overexpression mutants. With the wild type *S. atratus* SCSIO ZH16 as a control under the standard conditions for ilamycin production, overexpression mutants of *S. atratus* SCSIO ZH16:*ilaB* (Figure 3, trace v) were fermented. HPLC analysis of the fermentation media extracts revealed that the overall titer of ilamycins (**1**–**6**, Figure 1B) in *S. atratus* SCSIO ZH16:*ilaB* were enhanced 1.9-fold (7.45 mg/mL) (Figure 3B) as compared to those of in the wild type.

### 2.4. Identification of IlaJ/K as Type I ABC-Exporter Transporters Mediating the Secretion of Ilamycins 

*IlaJ* and *ilaK* were found in the middle of the ilamycin gene cluster and annotated as typical adenosine triphosphate (ATP)-binding cassette (ABC) transporter and ABC-2 type transporter, respectively, by BLAST analysis. The ABC transporter is a multi-domain protein consisting of four domains, two of which are transmembrane domains (TMD) and the other two are hydrophilic nucleotide-binding domains (NBD). The most common architecture of the ABC exporter protein is that of the so-called “half transporter,” which contains single TMD and NBD domains and has to interact with another “half transporter” to gain function [30]. In order to investigate the sequence characteristics of IlaJ and IlaK, sequence alignment and transmembrane characteristics were analyzed with online software such as ESPript [28] and TMHMM server 2.0 (Http://www.cbs.dtu.dk/services/TMHMM/). 

IlaJ/K showed high sequence similarity to other type I ABC-exporter transporters, such as DrrA/B [31], MtrA/B [32], TnrB2/B3 [33] and Atr29/30 [34], which involved in the secretion or self-defense for the producer of doxorubicin, mithramycin, tetronasin, and atratumycin, respectively. Conserved domains, such as Walker A (GXXGXGKS/T), Walker B (φφφφDE, φ represents hydroprobic amino acids), the ABC signature (L/YSGGQM), the Q-loop, and the switch region were presented in IlaJ (Appendix A), which suggest that IlaJ could function as an ATP-binding subunit. The sequence alignment results of IlaK with their homologues revealed that the typical “EAAxxxGxxxxxxxxxIxLP” motif [35] of the importer was not found in IlaK and their homologues (Appendix A). Moreover, transmembrane analyses showed that IlaJ does not contain any transmembrane domain, whereas IlaK has six transmembrane domains similar with those in DrrB, MtrB, TnrB3, and Atr30. These results indicate that IlaJ and IlaK may work together, forming functional exporter units to mediate the secretion of ilamycins or the self-defense of *S. atratus* SCSIO ZH16 to ilamycins.

To investigate the roles of IlaJ and IlaK in the biosynthesis of anti-TB ilamycins, three mutants, Δ*ilaJ*, Δ*ilaK*, and *ΔilaJK*, were constructed by the aforementioned method. Their phenotype and genotype were confirmed by Apr^R^/Kan^S^ and PCR analyses. PCR results are shown in Appendix A. Δ*ilaJ*, Δ*ilaK*, and Δ*ilajK* were fermented under the same conditions with the wild type as a control. To characterize the possible functions of IlaJ and Ilak in secretion of ilamycins or self-defense of *S. atratus* SCSIO ZH16 to ilamycins, the titers of ilamycins in both of the wild type and three mutants were quantified by HPLC analysis (Figure 4A) and standard curve (Appendix A). The quantified results revealed that the overall titers of ilamycins in Δ*ilaJ*, Δ*ilaK*, and Δ*ilaJK* were decreased to 69.5% (2.70 mg/mL), 59.9% (2.34 mg/mL), and 36.1% (1.42 mg/mL) to those of in the wild types (Figure 4B). Especially, the titers of ilamycin C_1_ and C_2_ in those three mutants were decreased by 50% to 86% compared to those of in the wild type (Figure 4B). Based on these findings, we propose that IlaJ and IlaK play roles in the biosynthesis of ilamycins, such as the secretion of ilamycins or self-defense of *S. atratus* SCSIO ZH16 to ilamycins. 

### 2.5. Heterologous Expression of the Ila Biosynthetic Gene Cluster (BGC) in Streptomyces Coelicolor M1152 

The biosynthetic gene cluster of ilamycins has been precisely characterized in our previous study [17]. To determine whether the *ila* BGC (*ca* 57 kb) can be successfully expressed in a *Streptomyces* model, such as *Streptomyces coelicolor* M1152. A PAC7A6 containing the proposed whole ilamycin gene cluster was introduced into *S. coelicolor* M1152 by tri-parental intergeneric conjugation [34,36,37]. The correction the exconjugants was verified by PCR amplification of the genes in the ilamycin cluster (Appendix A). HPLC analyses of the fermentation products of the exconjugants revealed several new peaks (Figure 3, trace vii) when compared with the wild type *S. coelicolor* M1152 (Figure 3, trace vi) under the same conditions. To confirm whether these new peaks in the fermentation products of the exconjugants were ilamycins, the ultraviolet spectra and the retention times of the new peaks were further analyzed, and the results reveal that they were identical to those seen in the wild type *S. atratus* SCSIO ZH16 (Figure 3, trace i), but the overall titers of ilamycins produced in *S. coelicolor* M1152/PAC7A6 were just one quarter of those in the wild type *S. atratus* SCSIO ZH16. These results indicate that the biosynthetic elements for the construction of ilamycins are all included in the PAC7A6 and ilamycins can be produced in heterologous *Streptomyces* models, such as *S. coelicolor* M1152. Furthermore, this result provides a reference for the discovery of other new complex cyclic peptides by heterologous expression.

## 3. Discussion

*Actinomycetes*, especially the genus *Streptomyces,* are very important sources of bioactive natural products [38,39]. However, the production of bioactive compounds is strictly controlled by many factors, such as the nutritional status, pH, temperature, dissolved oxygen, various environmental conditions, and regulators in their biosynthetic gene clusters [40,41]. Regulator genes and transport genes are important components of the secondary metabolite biosynthetic gene cluster, and they play important roles in changing the titers of these natural products. 

IlaA is a small protein of unknown function. Our results revealed that IlaA functioned as a negative regulator in the biosynthesis of ilamycins. IlaB belongs to the streptomycin biosynthesis operon regulator family. The typical representative of this kind regulator is StrR, a pathway-specific activator in the biosynthesis of streptomycin in *Streptomyces griseus* and *Streptomyces glaucescens*. The molecular mechanisms studies of StrR revealed that it was a pathway specific transcriptional activator protein with multiple recognition sites in the biosynthesis of streptomycin and was depended on by nine transcriptional units within the streptomycin biosynthesis gene cluster [42,43]. In light of the high identity and the structure similarity between StrR and IlaB, we propose that IlaB may have a similar regulatory mechanism in the biosynthesis of ilamycins to that of StrR in the biosynthesis of streptomycin. 

ATP-binding cassette (ABC) transporters belong to a large family of proteins that are widespread among living organisms. In bacteria, ABC transporters have a diverse range of functions, including an importer function mediating the uptake of nutrients and exporter function involving the secretion of various molecules, which are important for normal bacterial physiology [44,45]. The knockout of *atr29* and *atr30* led to the complete abolishment of atratumycin in *S. atratus* SCSIO ZH16NS-80S both in the pellet and the supernatant media [34]. In this study, the knockout of *ilaJ*, *ilaK*, and *ilaJK* led to an obvious decrease in the total yields of ilamycins, especially for the decrease of ilamycin C_1_ and C_2_, which indicated that they played important roles in the secretion of ilamycins or self-defense to ilamycins in *S. atratus* SCSIO ZH16. It is noteworthy that with the surge in genome sequencing of *Actinobacteria*, more ABC transporters will be identified in the biosynthetic gene clusters of natural products, and their roles in secretion or self-defense will be elucidated in secondary metabolite-producing strains [46]. ABC transporters can be used to construct high-yield strains or antibiotic-resistant model strains for heterologous expression of other BGCs.

Heterologous expression not only plays important roles in the discovery of novel natural products in the genomic era, but also provides a way to verify the integrity of the BGCs or to characterize the function of biosynthetic pathways in a genetically amenable host. Due to the progress of large DNA fragment cloning technology, the success rates of heterologous expression of larger biosynthetic gene clusters have significantly increased, as seen in the cases of atratumycin [34], neoabyssomicins [37], A201A [47], and grincamycin [48]. The successful expression of *ila* BGC in a heterologous host, which not only suggests the possibility of heterologous expression in the discovery of new natural products, but also confirms the feasibility of verifying the integrity of big gene clusters.

## 4. Conclusions

In summary, the small protein (annotated as a LysR family regulator) encoded by *ilaA* and the streptomycin biosynthesis operon regulator encoded by *ilaB* were demonstrated to be a negative and positive regulator, respectively, in the biosynthesis of ilamycin. Two overproducers were constructed by knockout of *ilaA* and overexpression of *ilaB* in *S. atratus* SCSIO ZH16. The secretion of ilamycins or self-defense of *S. atratus* SCSIO ZH16 to ilamycins by a pair of exporters encoded by *ilaJ* and *ilaK* was also characterized. Ilamycins were successfully produced in the *Streptomyces* model of *S. coelicolor* M1152. Our study on the regulatory and transport functions of IlaA/B/J/K not only sheds light on new characteristics of regulation, but also provides new genetic elements for synthetic biology. 

## 5. Material and Methods

### 5.1. Bacterial Strains, Plasmids, and Culture Conditions

Bacterial strains and plasmids used in this study are listed in Appendix A. *E. coli* strains, including DH5α and ΕΤ12567/pUZ8002, were cultivated at 37 °C in Luria-Bertani (LB) liquid medium or on LB agar. BW25113/pIJ790 was cultivated at 30 °C or 37 °C in super optimal broth (SOB) liquid medium, as necessary. *S. atratus* SCSIO ZH16 and its genetically engineered mutant strains were cultivated at 30 °C in P2 solid medium for sporulation and genetic manipulation [16]. When necessary, the medium was supplemented with apramycin 30 μg/mL, chloramphenicol 25 μg/mL, kanamycin 50 μg/mL, thiostrepton 12.5 μg/mL, or ampicillin 100 μg/mL.

### 5.2. General Genetic Manipulations and Reagents

General genetic manipulation of *Streptomyces* was carried out according to standard protocols. PCR amplification was performed on an Eppendorf Mastercycler^®^ EP gradient (Eppendorf, Germany) using high-fidelity Taq DNA polymerase purchased from TransGen Biotech Co. Ltd (Beijing, China). DNA fragments and PCR products were purified from agarose gels using a DNA Gel Extraction Kit (Omega). Primers were synthesized by Sangon Biotech Co. Ltd. Company (Shanghai, China). All DNA sequencing was performed by IGE Biotech Co. Ltd (Guangzhou, China). Restriction enzymes and T4 DNA ligase were purchased from New England Biolabs (Ipswich, MA, USA). 

### 5.3. Genomic Library Screening

The genomic cosmid library of *S. atratus* SCSIO ZH16 constructed with SuperCos1 has been described in our previous report [17]. Three pairs of primers (Appendix A) were designed and used to screen the genomic cosmid library by PCR. 

### 5.4. Construction of Genetic Mutants

All the mutant strains in this study were generated by homologous recombination according to the standard method [49]. For construction of the target mutants, the apramycin resistance cassette flanked by homologous arms with the target genes was amplified from the pIJ773 plasmid with Primer Star Taq enzymes (Takara) according to the manufacturer’s specifications. The fragments were purified from the agarose gel using a gel recycle kit according to the manufacturer’s specifications. The fragments were then electro-transformed into BW25113/pIJ790/plasmids. The disrupted plasmids were transformed into *E. coli* ET12567/pUZ8002 for conjugation. Interspecies conjugation was performed between *E. coli* ET12567/pUZ8002/disrupted plasmids and *S. atratus* SCSIO ZH16 on ISP4 solid medium supplemented with 20 mM MgSO_4_. Double-crossover mutants were identified through diagnostic PCR with corresponding primers listed in Appendix A. 

### 5.5. Complementation of the ΔilaB Mutant

For complementation of the *ilaB* mutant, the target gene *ilaB* was cloned into the vector pl646ATE under the control of *PermE**, which was digested with *Nde*I and *Spe*I, as reported by our group [34]. The correction of the target *ilaB* gene was verified by sequencing. The correctness of newly constructed complementary plasmids was verified by double enzyme digest, and the correct plasmids were transformed into *E. coli* ET12567/pUZ8002. Conjugation of *E. coli* ET 12567/ pUZ8002/ pl646ATE/*ilaB* with Δ*ilaB* mutant was then carried out. The correctness of conjugants was verified on the bases of the thiostrepton resistance phenotype and their genotypes.

For the construction of *ilaB* overexpression strain, the complemented plasmid, pl646ATE/*ilaB*, was introduced into *S. atratus* SCSIO ZH16 by conjugation with the aforementioned method. The correctness of the conjugants was verified on the bases of the thiostrepton resistance phenotype and their genotypes.

### 5.6. Heterologous Expression of the Biosynthetic Gene Cluster of Ilamycins

For heterologous expression of ilamycins, another genomic library was constructed with the pESAC13A vector by Bio S&T, Germany; this vector can adopt large fragments ca 100–120 kbp [34,37]. To identify colonies containing the whole gene cluster of ilamcycins, three pairs of primers were designed for screening the genetic library by PCR. The desired colonies were used for conjugation with *S. coelicolor* M1152 using the triparental conjugation methods described by Tao *et al.* [36]. Correctness of the conjugants was verified by PCR analysis.

### 5.7. Fermentation, Extraction, and Quantitative Analysis

For fermentation, the spores/mycelium of *S. atratus* SCSIO ZH16 and its genetically engineered mutant strains were inoculated into Am2ab liquid medium [17] and cultivated at 30 °C by shaking at 200 rpm for 7 days. All the genetically engineered mutants were extracted with two-fold volume of butanone. The extracts were evaporated in *vacuo* and re-dissolved in 1 mL methanol for HPLC analysis, as described previously [16]. HPLC conditions are as follows: solvent system (solvent A, 15% acetonitrile in water supplemented with 0.1% acetic acid; solvent B, 85% acetonitrile in water supplemented with 0.1% acetic acid); 20% B to 80% B (linear gradient, 0–20 min), 80% B to 100% B (linear gradient, 20–21.5 min), 100% B (21.5–27.0 min), 100% to 0% B (27.0–27.1 min), 0% B (27.1–30.0 min); flow rate was set as 1.0 mL/min. A reverse phase column (SB-C18, 5 μm, 4.6 × 150 mm) was used for analysis.

Quantitative analysis of ilamycin production was performed using GraphPad Prism 6 software. Ilamcyin yields were calculated using a calibration curve. 

## Figures and Tables

**Figure 1 marinedrugs-18-00216-f001:**
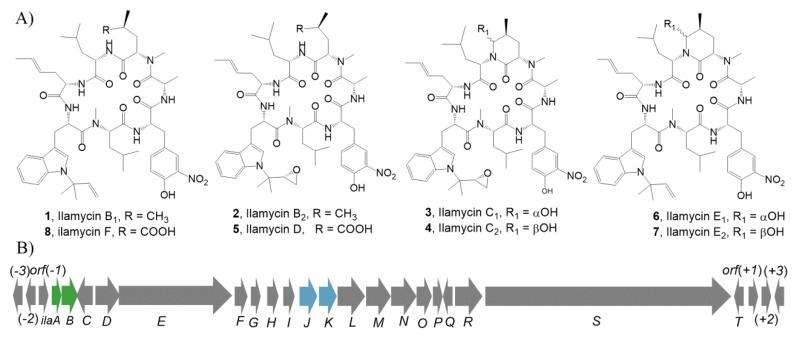
The structures and biosynthetic gene cluster of ilamycins. (**A**) The structures of ilamycins. (**B**) The genetic organization of ilamycin biosynthetic gene cluster, the genes marked with green and pale blue color encode the regulatory and transporter genes, respectively.

**Figure 2 marinedrugs-18-00216-f002:**
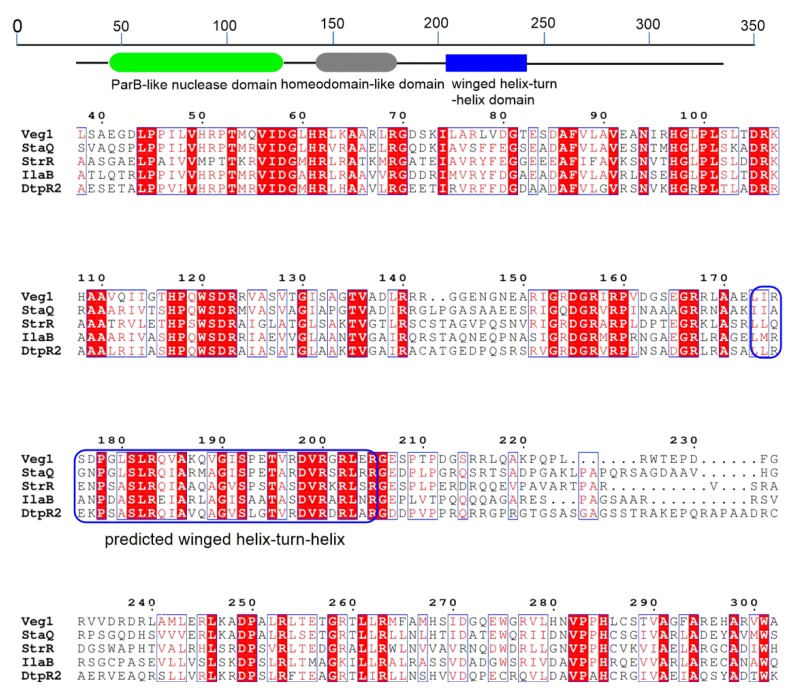
Domains in IlaB and primary sequence alignment of IlaB (ASX95224) with its homologues including StaQ (GenBank accession no. AAM80553) from *Streptomyces toyocaensis* NRRL 15009, StrR (GenBank accession no. BAG22760) from *Streptomyces griseus* subsp. *griseus* NBRC 13350, Veg1 (GenBank accession no. ACJ60943) from uncultured bacterium, DtpR2 (GenBank accession no. AJI44174) from *Saccharothix algeriensis*. The sequence encoding a predicted winged helix-turn-helix DNA binding motif is noted with a blue box.

**Figure 3 marinedrugs-18-00216-f003:**
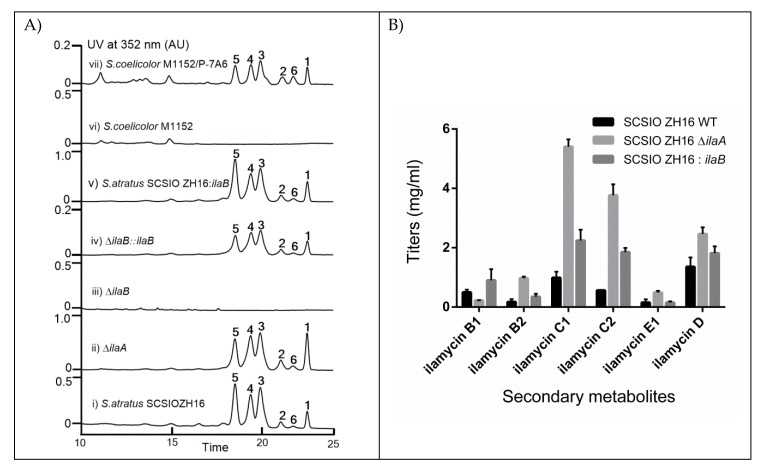
HPLC analysis of fermentation extracts and titers quantification. (**A**) HPLC analysis of fermentation of mutants and wild type. (i) *S. atratus* SCSIO ZH16, (ii) Δ*ilaA*, (iii) Δ*ilaB*, (iv) Δ*ilaB*::*ilaB*, (v) *S. atratus* SCSIO ZH16:*ilaB*, (vi) *Streptomyces coelicolor* M1152, (vii) *S. coelicolor* M1152-PAC7A6. (**B**) Comparative analysis of ilamycin yield between Δ*ilaA*, overexpression IlaB mutant and the wild type strain. The values are mean of three different clones.

**Figure 4 marinedrugs-18-00216-f004:**
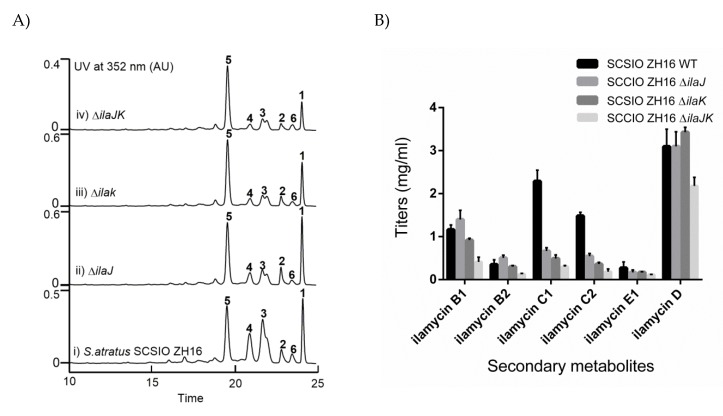
HPLC analysis of fermentation extracts and titers quantification. (**A**) (i) *S. atratus* SCSIO ZH16, (ii) Δ*ilaJ*, (iii) Δ*ilak*, (iv) Δ*ilaJk.* (**B**) Comparative analysis of ilamycin yield between Δ*ilaJ*, Δ*ilaK*, and Δ*ilaJK* mutants and the wild type strain. The values are mean of two different clones.

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
