# Peer review of "Characterization of Regulatory and Transporter Genes in the Biosynthesis of Anti-Tuberculosis Ilamycins and Production in a Heterologous Host"

_marinedrugs, 2020, doi:10.3390/md18040216_

Round 1
Reviewer 1 Report
The authors changed the article properly.
But I still have one additional concern described below.
I am sorry for making a bad mistake about the homolog of IlaB.
But anyway, I confirmed the BLAST search again and found that IlaB seemed to be the homolog of StrR which is a regulator of streptomycin biosynthesis.
If the accession number of IlaB is ASX95224 (this is not actually mentioned in the text), the identity of IlaB with StrR is 52% comparable to other regulators in the text.
StrR should be included in the manuscript and some reference described below should be included.
I also attached the three parent alignment of StaQ (AAM80553), IlaB (ASX95224) and StrR (BAG22760).
I appreciate if authors can include my additional suggestion.
streptomycin biosynthesis operon regulator [Streptomyces griseus subsp. griseus NBRC 13350]
Sequence ID: BAG22760.1Length: 350Number of Matches: 1
Score Expect Method Identities Positives Gaps
267 bits(682) 4e-88 Compositional matrix adjust. 159/307(52%) 194/307(63%) 8/307(2%)
Query 39 SAIECVSVAALTTSDTPRSGGEDEEHVQLLATLQTRLPPIVVHRPTMRVIDGAHRLRAAV 98
SA+ V V +L SD+PRS GED EH++ LA LP IVV T RVIDG HRLRA
Sbjct 31 SAVTMVPVESLLPSDSPRSAGEDVEHIRTLAASGAELPAIVVMPTTKRVIDGMHRLRATK 90
Query 99 VRGDDRIMVRYFDGAEADAFVLAVRLNSEHGLPLSLTDRKAAAARIVASHPQWSDRRIAE 158
+RG I VRYF+G E +AF+ AV+ N HGLPLSL DRKAAA R++ +HP WSDR I
Sbjct 91 MRGATEIAVRYFEGGEEEAFIFAVKSNVTHGLPLSLDDRKAAATRVLETHPSWSDRAIGL 150
Query 159 VVGLAANTVGAIRQRSTAQNEQPNASIGRDGRMRPRNGAEGRLRAGELMRANPDASLREI 218
GL+A TVG +R STA Q N IGRDGR RP + EGR A L++ NP ASLR+I
Sbjct 151 ATGLSAKTVGTLRSCSTAGVPQSNVRIGRDGRARPLDPTEGRKLASRLLQENPSASLRQI 210
Query 219 ARLAGISAATASDVRARLNRGE-PLVTPQQQQAGARESP--AGSAARRSVRSGCPASE-V 274
A AG+S +TASDVR RL+RGE PL +QQ E P A + AR S G A V
Sbjct 211 AAQAGVSPSTASDVRKRLSRGESPLPERDRQQ----EVPAVARTPARVSRADGSWAPHTV 266
Query 275 LLVSLSKDPSLRLTMAGKILLRALRASSVDADGWSRIVDAVPPHRQEVVARLARECANAW 334
L LS+DPS+RLT G+ LLR L +V W R++ VPPH +V+A LAR CA+ W
Sbjct 267 ALRHLSRDPSVRLTEDGRALLRWLNVVAVRNQDWDRLLGNVPPHCVKVIAELARGCADIW 326
Query 335 QLLADRL 341
+A+ L
Sbjct 327 HRVAEEL 333
Properties of C-terminal truncated derivatives of the activator, StrR, of the streptomycin biosynthesis in Streptomyces griseus.
Thamm S, Distler J.
FEMS Microbiol Lett. 1997 Apr 15;149(2):265-72.
PMID: 9141668 Free Article
The regulator of streptomycin gene expression, StrR, of Streptomyces griseus is a DNA binding activator protein with multiple recognition sites.
Retzlaff L, Distler J.
Mol Microbiol. 1995 Oct;18(1):151-62.
Transcriptional control by A-factor of strR, the pathway-specific transcriptional activator for streptomycin biosynthesis in Streptomyces griseus.
Tomono A, Tsai Y, Yamazaki H, Ohnishi Y, Horinouchi S.
J Bacteriol. 2005 Aug;187(16):5595-604.

Author Response
Reviewer 1#
The authors changed the article properly. But I still have one additional concern described below.
Response: We appreciate reviewer #1 for the positive comments on our revised manuscript.
I am sorry for making a bad mistake about the homolog of IlaB. But anyway, I confirmed the BLAST search again and found that IlaB seemed to be the homolog of StrR which is a regulator of streptomycin biosynthesis. If the accession number of IlaB is ASX95224 (this is not actually mentioned in the text), the identity of IlaB with StrR [streptomycin biosynthesis operon regulator, Streptomyces griseus subsp. griseus NBRC 13350] is 52% comparable to other regulators in the text. StrR should be included in the manuscript and some reference described below should be included.I also attached the three parent alignment of StaQ (AAM80553), IlaB (ASX95224) and StrR (BAG22760). I appreciate if authors can include my additional suggestion. [Thamm S, Distler J. Properties of C-terminal truncated derivatives of the activator, StrR, of the streptomycin biosynthesis in Streptomyces griseus. FEMS Microbiol Lett. 1997, 149(2):265-72; Retzlaff L, Distler J. The regulator of streptomycin gene expression, StrR, of Streptomyces griseus is a DNA binding activator protein with multiple recognition sites. Mol Microbiol. 1995, 18(1):151-62; Tomono A, Tsai Y, Yamazaki H, Ohnishi Y, Horinouchi S. Transcriptional control by A-factor of strR, the pathway-specific transcriptional activator for streptomycin biosynthesis in Streptomyces griseus. J Bacteriol. 2005, 187(16):5595-604.].
Response: We appreciate reviewer #1 for this excellent suggestion. ASX 95224, the accession number of IlaB, has been added in the annotation of Figure 2. StrR, a homolog of IlaB, has been included in the sequence alignment in Figure 2 and added in the main text in Line 80 (Page 2). In addition, the discussion on StrR and IlaB to read to “IlaB belongs to the streptomycin biosynthesis operon regulator family. The typical representative of this kind regulator is StrR, a pathway-specific activator in the biosynthesis of streptomycin in Streptomyces griseus and Streptomyces glaucescens. The molecular mechanisms studies of StrR revealed that it was a pathway specific transcriptional activator protein with multiple recognition sites in the biosynthesis of streptomycin and was depended on by nine transcriptional units within the streptomycin biosynthesis gene cluster [42,43]. In light of the high identity and the structure similarity between StrR and IlaB, we propose that IlaB may have a similar regulatory mechanism in the biosynthesis of ilamycins to that of StrR in the biosynthesis of streptomycin”, and the references on StrR have been added in the main text (Page 6) and the reference list.
Reviewer 2 Report
In this second version of the paper, authors have improved the part corresponding to quantification of ilamycins production in WT, ilaJ, and ilaK mutants. Their results now show more convincingly that these genes may code for parts of the export apparatus.
On the other hand, authors have not performed additional experimentation requested by one or both reviewers in order support their conclusions on the regulatory role of IlaA and IlaB proteins. Authors´rebuttals are not convincing:
1) Both reviewers asked to show expression of ila biosynthetic genes in the postulated regulatory mutants but this has not been done. Particularly, although IlaA was initially annotated as a LysR-type regulator, its function is really unknown, as authors admit themselves (lns. 71-73). Alright, IlaB bears an HTH DNA-binding domain, therefore, it is presumably a transcriptional regulator (activator, according to the ilaB mutant phenotype). However, in my opinion authors should show ilamycin biosynthetic genes expression in the mutants.
2) In my previous 2nd comment, I was concerned about the IlaA deletion mutant phenotype being due to ectopic expression of the putative activator ilaB, rather than to lack of the IlaA putative repressing activity, not concerned about a possible polar effect of the ilaA mutation, which would have a lack of expression phenotype due to the absence of IlaB. I requested to show expression of ilaB in WT and ilaA deletion mutant. This has not been done and the rebuttal argument is not satisfactory. The different ilamycin production levels in the trans-complemented ilaB deletion mutant and in the wt expressing ilaB from PermE does not indicate anything about IlaA, which is present in both strains, it just indicate that IlaB is produced to higher levels from the chromosomal gene copy than from the plasmid one (complementation of the ilaB deletion mutant is poor, with lower levels of ilamycins production than WT or WT with ilaB expressed fom PermE).
Indeed, authors neither performed the ilaA mutant trans-complementation requested by reviewer 1, which would have shed light into this issue.
3) My previous third comment may have been misinterpreted. I was asking whether the IlaB activator could be dispensable for expression of ila biosynthetic genes in the absence of the putative repressor IlA, i. e. in an ilaA deletion background. Because of that I requested the double ilaAB deletion mutant.
Author Response
Reviewer 2#
In this second version of the paper, authors have improved the part corresponding to quantification of ilamycins production in WT, ilaJ, and ilaK mutants. Their results now show more convincingly that these genes may code for parts of the export apparatus.
Response: We appreciate reviewer #2 for the positive comments on our revised manuscript.
On the other hand, authors have not performed additional experimentation requested by one or both reviewers in order support their conclusions on the regulatory role of IlaA and IlaB proteins. Authors´rebuttals are not convincing:
1) Both reviewers asked to show expression of ila biosynthetic genes in the postulated regulatory mutants but this has not been done. Particularly, although IlaA was initially annotated as a LysR-type regulator, its function is really unknown, as authors admit themselves (lns. 71-73). Alright, IlaB bears an HTH DNA-binding domain, therefore, it is presumably a transcriptional regulator (activator, according to the ilaB mutant phenotype). However, in my opinion authors should show ilamycin biosynthetic genes expression in the mutants.
Response: We appreciate reviewer #2 for this point. Suggested by reviewer #1, the bioinformatics analysis and BlAST analysis of IlaB showed that IlaB was a homologue of StrR with 52% identity, which was a pathway specific transcriptional activator protein with multiple recognition sites in the biosynthesis of streptomycin and the whole transcription of streptomycin gene cluster depended on StrR [Retzlaff L, Distler J. Mol Microbiol. 1995, 18(1):151-62; Tomono A, Tsai Y, Yamazaki H, Ohnishi Y, Horinouchi S. J Bacteriol. 2005, 187(16):5595-604.]. Moreover, our further in vivo inactivation of ilaB, in trans gene complementation of DilaB mutant and the overexpression of ilaB in S. atratus SCSIO ZH16 revealed that IlaB is essential in the biosynthesis of ilamycins in S. atratus SCSIO ZH16 and IlaB is an transcriptional activator in the biosynthesis of ilamycins. We proposed that as a highly homologous protein of StrR, IlaB may have the same regulatory mechanism as StrR in the biosynthesis of ilamycins. In other words, the six transcriptional units in the biosynthetic gene cluster of ilamycin maybe all dependent on IlaB. In addition, the discussion on StrR and IlaB, and the references on StrR have been added in the main text (Page 6). Due to the severe situation of COVID-19, all the labs in China have been closed and the experiment can’t be carried out.
2) In my previous 2nd comment, I was concerned about the IlaA deletion mutant phenotype being due to ectopic expression of the putative activator ilaB, rather than to lack of the IlaA putative repressing activity, not concerned about a possible polar effect of the ilaA mutation, which would have a lack of expression phenotype due to the absence of IlaB. I requested to show expression of ilaB in WT and ilaA deletion mutant. This has not been done and the rebuttal argument is not satisfactory. The different ilamycin production levels in the trans-complemented ilaB deletion mutant and in the wt expressing ilaB from PermE does not indicate anything about IlaA, which is present in both strains, it just indicate that IlaB is produced to higher levels from the chromosomal gene copy than from the plasmid one (complementation of the ilaB deletion mutant is poor, with lower levels of ilamycins production than WT or WT with ilaB expressed fom PermE). Indeed, authors neither performed the ilaA mutant trans-complementation requested by reviewer 1, which would have shed light into this issue.
Response: We thank reviewer #2 for this point. From the molecular mechanism study of StrR, a highly homologue of IlaB, we knew that the transcriptional expression of strR was controlled by A-factor, which is a representative of the g-butyrolactone auto-regulator controlling secondary metabolism, morphological development, or both particularly in the genus Streptomyces. AdpA, one key transcriptional activator in the A-factor signal transduction cascade, assisted RNA polymerase in forming an open complex at an appropriated position for the transcriptional initiation of strR [Horinouchi, S. Front. Biosci. 2002, 7, d2045-d2057; Ohnishi Y., Seo J.-W., Horinouchi, S. FEMS Microbiol. Lett. 2002, 216, 1-7]. From those results of StrR, we propose that IlaB may have a similar regulation mechanism to that of StrR and the expression of IlaA and IlaB is independent of each other. In this manuscript, we mainly focused on the functions of four genes in ilamycin biosynthetic gene cluster (ila BGC) including two regulatory genes (ilaA and ilaB) and two transporter genes (ilaJ and ilaK) and the successful construction of two overexpression mutants of ilamycins, as well as the successful heterologous expression of ila BGC in S. coelicolor M1152. Whether the titer increase of ilamycins in DilaA mutant is caused by the overexpression of IlaB will be verified in future experiments.
3) My previous third comment may have been misinterpreted. I was asking whether the IlaB activator could be dispensable for expression of ila biosynthetic genes in the absence of the putative repressor IlaA, i. e. in an ilaA deletion background. Because of that I requested the double ilaAB deletion mutant.
Response: We thank reviewer #2 for this suggestion. Just like the answer to the question 2, combined the molecular regulatory mechanism study of StrR with our bioinformatics analysis, in vivo inactivation, in trans complementation of ilaB and in vivo inactivation of ilaA, we proposed that the expression of ilaA and ilaB is independent of each other, and the IlaB, a highly homologue of StrR, maybe control by the A-factor similar to that of StrR. The expression of the whole ila gene cluster may depended on IlaB. The detailed mechanism whether the IlaB activator could be dispensable for the expression of ila biosynthetic genes in the absence of the putative repressor IlaA will be characterized in the future.
Round 2
Reviewer 2 Report
Authors have comented my concerns in more detail and I understand that authors cannot perform further experimentation due to the exceptional situation caused by the COVID-19 pandemic.
This manuscript is a resubmission of an earlier submission. The following is a list of the peer review reports and author responses from that submission.
Round 1
Reviewer 1 Report
In this manuscript, the authors analyzed regulators and transporters in ilamycin biosynthesis. The ilamycin biosynthetic gene cluster contains two putative regulators ilaA and ilaB. The ilaA mutant produced higher amount of ilamycin while ilaB mutant did not produce any ilamycin. These results indicated that IlaA and IlaB are negative and positive regulators of ilamycin. Next, they focused on ilaJ and ilaK encoding transporters. The inactivation of these genes seemed to be slightly affecting the accumulation of ilamycins in the cells. Finally, the authors introduced the whole gene cluster into Streptomyces coelicolor and established the heterologous expression. The manuscript is written in good form and most of the part is understandable. However, I have several concerns on this research.
First, the authors are claiming that the IlaB is involved in a unique family of regulator and is the first regulator to be analyzed in this family. However, there is a paper describing the analysis of DptR2 (https://www.ncbi.nlm.nih.gov/pubmed/24768321) which is described as a IlaB homolog in this paper. Therefore, the authors should refer to this paper and should tone down their claim. In addition, this paper describes that the DptR2 is not controlling the expression of genes involved daptomycin biosynthesis (such as dptA) and should not be a pathway specific regulator. Considering the identity between IlaB and DptR2 (52%), it might be same for IlaB. To address this concern, I think the authors should check the expression of ilamycin biosynthetic genes in wt, ilaB mutant and complementation strain.
I also recommend to carry out complementation experiment for ilaA mutant.
Although the authors are providing actual yield and p value for ilaA and ilaB mutant, such data is not provide for ilaJ and ilaK mutants. The authors should provide similar experimental data for these mutants as well.
Finally, the authors succeeded in heterologous production of ilamycin in S. coeliolor. Although the importance of the data, the authors did not mentioned about the yield in the heterologous strain. Because most of the reader should be interested in this point, the yield of ilamycins in this heterologous host should be also described.
Reviewer 2 Report
Authors have extended their previous work on the ilamycins biosynthetic cluster and shown the function of four genes, two of them being involved in ilamycins export and two others regulating ilamycins biosynthesis in opposite ways. Authors have constructed 4 knockout mutants and analised their genotypes and phenotypes. These mutants bear a deletion of a particular ila gene but also an insertion of the apramycin resistance gene flanked by FRT sequences, which might affect expression levels of the downstream ila genes of the same operon.
Please, see my comments below:
1) Although authors have shown the effects of regulatory genes on ilamycins biosynthesis, I miss a measurement of actual gene transcription, such as RT-qPCR, of at least one of the ila biosynthetic genes, for instance, ilaD.
2) Part of the ilaA gene in the ilaA deletion mutant is actually replaced by the aac(3)IV gene flanked by the FRT sequences. Therefore, expression of the downstream ilaB gene could be affected by transcriptional readthrough from aac(3)IV. Since ectopic expression of ilaB from the ermE promoter leads to increased ilamycins production, how do authors know that the increasing effect in ilamycins biosynthesis of the ilaA deletion is not due to over-transcription of ilaB? RT-qPCR of ilaB in the WT and ilaA deletion mutant strains should show whether the level of ilaB transcription has been altered.
3) If IlaA actually worked as a repressor, it would be interesting to know whether IlaB is required for ila gene expression in the absence of IlaA. In other words, what the ila genes expression levels and ilamycins production would be in a ilaA ilaB double mutant.
4) It is not obvious to me that there are fewer amounts of ilamycins in the supernatants of of the ilaJ or ilaK mutant cultures, as stated in Results and in Discussion. Actually, from the HPLC data shown in Supplementary Figure 7, it looks like there are even higher amounts of ilamycins in the case of the mutants. The same can be observed in the pellet fractions. Convincing quantitative data of total production of ilamycins and their fractionation in pellet and supernatant fractions of WT, ilaJ and ilaK strains should be provided in order to establish the role of IlaJ and IlaK as ilamycin export proteins. These data should be presented in the main text and not as Supplementary material.
5) Actually, what is observed in Suppl. Fig. 7 is that ilaJ and ilaK mutant strains accumulate an additional compound that eluted right after peak 3. Do authors have an explanation for this? Could this be due to an intermediate accumulation due to overexpression of some genes downstream ilaK? Please, mind that the apramycin resistance gene is substituting the deleted genes.
6) Pg. 5, ln. 154-155: Authors state “the typical “EAAxxxGxxxxxxxxxIxLP” motif found in exporter proteins was not found in IlaK”. Is not this an indication that there is something missing in this protein couple IlaJ-IlaK?
7) In my opinion, the manuscript would benefit from being copy-edited by a native English speaker.
Other minor comments:
8) Current Figure 1B is not cited in the main text. I think it should be specifically cited in Introduction, lns. 48-49. By the way, panels in this figure should then be exchanged. For instance, in lane 49:
“…, the structure of six of them being shown in Figure 1A, were isolated by our group…”
9) Why ilamycins E2 and F are not shown in Figure 1B?
10) Figure 2: “nuclease” instead of nuslease
11) Pg. 3, ln. 89: I do not think the HTH domain is “…near the C-terminal (end)…” It is in the second half of IlaB, but in the central part.
12) Pg. 5, ln. 166. There is no Figure 4.
13) Ln., 199: …in the C-terminal region…, not “downstream”
Ln., 200: …in the N-terminal region…, not “upstream”
14) Ln. 232 produced instead of “expressed”